# Advantages in Using Colour Calibration for Orthophoto Reconstruction

**DOI:** 10.3390/s22176490

**Published:** 2022-08-29

**Authors:** Francesco Tocci, Simone Figorilli, Simone Vasta, Simona Violino, Federico Pallottino, Luciano Ortenzi, Corrado Costa

**Affiliations:** Consiglio per la Ricerca in Agricoltura e L’analisi Dell’economia Agraria (CREA), Centro di Ricerca Ingegneria e Trasformazioni Agroalimentari, Via della Pascolare 16, Monterotondo, 00015 Rome, Italy

**Keywords:** UAV, UGV, drones, mapping, precision agriculture, environmental monitoring

## Abstract

UAVs are sensor platforms increasingly used in precision agriculture, especially for crop and environmental monitoring using photogrammetry. In this work, light drone flights were performed on three consecutive days (with different weather conditions) on an experimental agricultural field to evaluate the photogrammetric performances due to colour calibration. Thirty random reconstructions from the three days and six different areas of the field were performed. The results showed that calibrated orthophotos appeared greener and brighter than the uncalibrated ones, better representing the actual colours of the scene. Parameter reporting errors were always lower in the calibrated reconstructions and the other quantitative parameters were always lower in the non-calibrated ones, in particular, significant differences were observed in the percentage of camera stations on the total number of images and the reprojection error. The results obtained showed that it is possible to obtain better orthophotos, by means of a calibration algorithm, to rectify the atmospheric conditions that affect the image obtained. This proposed colour calibration protocol could be useful when integrated into robotic platforms and sensors for the exploration and monitoring of different environments.

## 1. Introduction

For many years, mapping crops through remote or proximal sensing using unmanned aerial or ground vehicles (UAV or UGV) represents a crucial technology used in Precision Agriculture and Smart Farming [1]. Indeed, these technologies are commonly used for monitoring crop fields and providing effective solutions for precise farm management [2]. These powerful tools aid farmers with real-time data and image mapping strategies.

Typically, UAVs in precision agriculture are used for outdoor environments (rarely indoors, such as greenhouses) [3] where Global Navigation Satellite System (GNSS) access is available to provide more reliable control of the UAV in both manual and autonomous flights [4]. In fact, when using the GNSS and dedicated apps, it is possible to plan specific missions while optimising flight duration and sensor acquisition from drones in order to monitor natural and agricultural resources and take proper actions [5]. UAVs represent the first step toward Digital Agriculture, allowing aerial data to be collected and processed using specific software. Among their usages, one is the assessment of tillage quality parameters, with the goal of evaluating the field performance of agricultural equipment. This allows farmers to compare both implements and tillage methods, helping them to choose those most useful for their agronomic needs and thus achieve both good work quality and energy savings [6]. Moreover, in agriculture, drones are used for biomass monitoring, crop vigour and growth monitoring, food quality assessment, pathogens and pest identifications (for site-specific application of agrochemicals), harvesting and logistics optimisation, and many others [7]. These applications involve the processing of images acquired by sensors embedded in the drone. The three types of precision agriculture applications areas can be distinguished based on multispectral, thermal, or RGB camera sensors in the drones, with the last being widely available due to the low cost of the sensors [8]. Remotely acquired images are used to obtain orthoimages, Digital Elevation Models (DEM), and Digital Surface Models (DSM) [9,10]. The orthophotos contain information about the original data and the geometry corresponding to the maps.

In detail, orthophotos represent high-precision data collection for both precision agriculture and forest management based on orthogonal projections. To obtain an orthophoto, one must have an image, usually a Digital Surface Model (DSM) [11].

After the flight is performed by drones or other carriers, it is important to verify how the orthophoto is generated. In fact, aerial images obtained by UAVs can be affected by several factors (such as atmospheric refraction, terrain curvature, and perspective projection) that can lead to image deformation with regard to pixels [12]. Moreover, during data acquisition, exposure parameters and weather conditions are important and can greatly affect the final image.

As reported by Wierzbicki et al. [13], meteorological conditions reduce the radiometric quality of images. Consequently, low radiometric quality affects the density and accuracy of the point clouds and digital terrain model and the accuracy of the orthophoto. For this reason, if the flight is to be performed under overcast or rainy conditions, the flight parameters should be properly customised in order to achieve satisfactory processing accuracy. In fact, when capturing images in bad weather conditions, the required DSM should be reduced by 25–30% to obtain comparably accurate results as when using photos taken in good conditions. Consequently, for most systems, this will result in a reduction in flight altitude.

Generally, orthophotos obtained by a UAV under various atmospheric and illumination conditions have a number of defects. These defects arise directly from the acquisition method and have an impact on the quality of the photogrammetric product. In the work by Burdziakowski et al. [14], the use of UAVs during the darkest hours of the day was discussed (currently, photogrammetry in these lighting conditions is a topic not yet covered in depth in the bibliography). However, it has been shown that the dark period of the day promotes background elimination and light source extraction, eliminating any defects [14]. As reported by Wierzbicki et al. [15], orthophotos from UAVs can have radiometric distortions and inhomogeneities, such as variations in the radiation source (the sun), terrain relief, the directionality of reflection or emission of radiation from the Earth’s surface, absorption, and scattering in the atmosphere. Weather conditions (clouds and precipitation) and lighting conditions during image acquisition and camera sensor sensitivity are also important. Low radiometric image quality results in deterioration of the final accuracy of photogrammetric and remote sensing products.

To overcome this problem, and thus improve the quality of orthophotos, innovative photogrammetric techniques are being developed [11]. One method for orthophoto reconstruction is the MAGO (Adaptive Mesh for Orthophoto Reconstruction) approach. The input data of the MAGO procedure are the image (including its orientation parameters), the user-defined orthophoto plane, and a point cloud defining the object. Using such an approach, each pixel of the image is projected onto the orthophoto plane through an iterative process (which builds an adaptive mesh) [11].

Colorimetric calibration prior to photo/image interpretation is gaining increasing attention worldwide in many diverse disciplines such as biology since the colour is both a phenetic characteristic of organisms and a feature of the environmental space occupied by organisms [16,17]; however, real-world applications of such approaches in the agricultural sector are still scarce [3]. Moreover, different lighting conditions, such as different day times or a clear/cloudy sky, result in different light colour temperatures, and, thus, substantial differences in the acquired images. The acquisition environment can change during acquisition with a potential flight failure due to excessive colour or exposure differences among images. In any of the mentioned cases, it is desirable to try to reduce as many potential problems as possible to increase the reconstruction performance.

In this regard, the purpose of this work is to verify the advantages of an innovative colour calibration algorithm for UAV images used for orthophoto reconstruction under different environmental conditions. This work represents a breakthrough because, although colorimetric calibration can help in orthophoto reconstruction in the presence of different lighting and weather conditions, currently no studies on this topic have been found in the present literature.

## 2. Materials and Methods

### 2.1. Experimental Data

On three consecutive days (14–16 January 2020), several flights (18) with a UAV DJI Spark (DJI SZ DJI Technology Co., Ltd., Shenzhen, China) were conducted on an experimental agricultural field of around 2.5 hectares located in Monterotondo Scalo (Central Italy; N 42.103145425494446, E 12.628124559510711). The surface is composed of a plain field set aside. The field was divided into six zones (three horizontal and three longitudinal; Figure 1A,B) The three days had different weather conditions and different daytimes as reported in Table 1.

### 2.2. UAV Image Acquisition and Orthoimage Reconstruction

The UAV field reconstruction was performed at a height of 20 m above the ground starting from an overfly. The UAV characteristics are reported in Table 2.

The flight of the UAV was planned with the software Mission Planner software [18]. Mission Planner allows for the flight design to obey several prescriptions related to the surface to be acquired, such as sidelap and overlap, ground sampling distance (GSD), total flight time, and image acquisition synchronisation. The mission was exported to the application Litchi for Android used to flight control and pilot the UAV. The application loads waypoints through csv files for a predefined mission flight.

The UAV digital camera (DJI Spark—model FC1102) collected still images every 3.34 s using a shutter speed of 1/533 s and sensor sensitivity of ISO 100. The camera specifications are described in Table 1. Images were collected using the UAV flying with a velocity of 2 m/s at 22 m above ground level (AGL), obtaining a GSD (Ground Sample Distance) equal to 0.750 cm/pixel. The timing of the shots was configured to obtain an overlap between photos of 70% of the image and a sidelap of 68% of the image. For every zone, the drone covered a surface of 9802 m^2^ on average (a total area of around 3 hectares). For all six zones, the UAV compass was heading 129° SE. The images were acquired with uneven light during the three days of flying, under the influence of weather conditions and sun position during the day. The total flight time (including take-off and landing) was around 12 min.

At the beginning of each flight, an image colour checker GretagMacbeth (24 patches) (X-Rite, Grand Rapids, MI, USA) was acquired for the colour calibration. This image serves for the posteriori colour calibration. The a priori knowledge of the colour checker patches values allowed the colour calibration of all the images following the thin-plate spline interpolation function [17] in the RGBs space values following a defined procedure developed by Menesatti et al. [17] using a MATLAB (MathWorks Inc., Natick, MA, USA) procedure. The thin-plate spline refers to the bending of a thin metal plate. The bending occurs in the z-direction, orthogonal to the plane. In the coordinate transformation, the lifting of the plate is interpreted as a displacement of the x or y coordinates within the plane. In 2D cases, given a set of K corresponding points, the TPS deformation is described by 2(K + 3) parameters (six global affine motion parameters and 2K coefficients for control point correspondences). The TPS has a closed form and, consequently, a modification must be performed to produce interpolation functions for three-dimensional thin-plate splines. The thin-plate spline is a plane-to-plane map that maps each reference point to its corresponding one. This method allows minimising the effects of the illuminant, camera characteristics, and settings by measuring the ColorChecker‘s RGB coordinates within the acquired images and transforming them into the known reference coordinates of the ColorChecker.

After image acquisition, the orthophotos were reconstructed using the software “Agisoft Metashape Professional version 1.6.6″ according to the following steps: project creation; identification of the same region (bounding box) and align photo with GPS information inside the image; settings of GCP (Ground Control Points), optimising the camera alignment without GPS inside the image and with GCP; build a dense cloud, build mesh, build texture, build tiled model, and build orthomosaic. The final step consists of exporting the orthophoto in tiff files with LZW compression.

A comparison between the orthophotos obtained with calibrated and non-calibrated images was conducted. Five Ground Control Points have been used to reconstruct the orthophoto, and the other four (Quality Control Point—checkpoints) were used to check the quality of the reconstruction. The GCP and checkpoint are always the same for all the calibrated and non-calibrated reconstructions to avoid influences on the results.

A total of 30 orthophotos were obtained: 15 from calibrated images and 15 from non-calibrated images; among which, 3 were from the reconstruction of the zones of each day and 12 randomly combined the six zones over the three days of the experiment. The combination of the zones was the same between calibrated and non-calibrated orthophoto reconstructions.

A series of parameters were obtained after orthophoto reconstruction: Number of images used for the reconstruction, Number of tie points, Number of projections, Reprojection error, Control points Total Root Mean Square Error (RMSE), and Check points Total RMSE.

Percentage Camera stations/Nimg: represents the percentage of images used for the reconstruction with respect to the total number of images available; this parameter indicates how many “good” images serve for the reconstruction.

A tie point is a point in a digital image or aerial photograph that represents the same location in an adjacent image or aerial photograph; the larger the number of tie points, the more accurate the reconstruction.

The number of projections represents the total number of projections projected from all overlapping images to compute and construct all the matched points. The number of projections is correlated to the number of points successfully matched [19].

The reprojection error is a geometric error corresponding to the image distance between a projected point and a measured one. It is used to quantify how closely an estimate of a 3D point recreates the point’s true projection.

Control points Total RMSE represents the total error of the five GCP positioning.

Check points Total RMSE represents the total error of the four checkpoints positioning.

To make the spatial quantitative comparison between orthophotos, photos were superimposed using the nine ground control points (landmarks) on each of the 30 images. A consensus configuration (i.e., average shape) was calculated [20]. The software TPSsuper [21] allowed us to unwarp the images to the consensus shape (target shape). The orthophotos, in this way, were aligned to landmark locations and superimposed.

After the orthophoto superimposition, the central part of the agricultural (Figure 1C) field was sampled with 300 random points. The mean value of the 8 × 8 neighbour pixels was extracted, giving rise to a population of 30 average pixels. The R, G, and B distributions over the 30 images were then considered and for each channel, the standard deviations were extracted. To obtain a variability index of an average random pixel over the 30 images, for each random point extracted, the Root Mean Square (RMS) of the standard deviations in the three R, G, and B channels was calculated. The 300 values of the RMS standard deviations (SD) associated with the three channels of the calibrated and non-calibrated orthophoto were compared by means of a paired Wilcoxon test for non-parametric distributions.

## 3. Results

Figure 2 shows an example of a calibrated and non-calibrated orthophoto reconstructed randomly combining the same six zones over the three days of the experiment. Figure 3 shows an example of calibrated and non-calibrated images including a GCP. It is clearly possible to observe how the calibrated orthophoto is greener and brighter than the non-calibrated one, representing the real colours of the scene. Additionally, in terms of the homogeneity of the orthophoto, the calibrated one is much more homogeneous, while in the non-calibrated one, the different zones owing to different flights are easier to identify.

Figure 4 shows the comparison between a calibrated and non-calibrated orthophoto reconstructed with images of the same day (day 1) and randomly combining the same six zones over the three days of the experiment. It is possible to observe how the image of the same day is more homogeneous (i.e., the six zones are not clearly visible) but less green and brighter than the calibrated one (Figure 2A) representing the real colours of the scene.

Differences between the parameters obtained after orthophoto reconstruction from calibrated and non-calibrated reconstructions are reported in Table 3. It is possible to observe how parameter reporting errors are always lower in the calibrated reconstructions and the other quantitative parameters are always lower in the non-calibrated ones. Considering that the paired comparison (Wilcoxon test) was only conducted on 15 reconstructions, it is still possible to observe significant (*p* < 0.05) differences in the percentage of camera stations on the total number of images and the Reprojection error.

After the orthophoto superimposition, the central part of the agricultural (Figure 1C) field was sampled with 300 random points in order to evaluate the comparison considering the colour homogeneity. The mean value of the 8 × 8 neighbour pixels was extracted; giving rise to a population average of 30 pixels. The R, G, and B distributions over the 30 images were then considered and the standard deviations were extracted for each channel. The variability index of an average random pixel over the 30 images, for each random point extracted, was extracted through the Root Mean Square (RMS) of the standard deviations in the three R, G, and B channels (RMS SD-RGB). The 300 values of the RMS standard deviations (SD) associated with the three channels of the calibrated and non-calibrated orthophoto were compared by means of a paired Wilcoxon test for non-parametric distributions which reported a highly significant difference (*p* < 0.0001) between the calibrated and non-calibrated images (Table 3). Reconstructions with the calibrated images showed a significantly lower RMS with respect to non-calibrated ones (i.e., the calibrated orthophoto showed to have a much more homogeneous colour rendering).

## 4. Discussion

As demonstrated by Wierzbicki et al. [15], atmospheric conditions, especially radiometric inhomogeneities, such as variations in the source of solar radiation and directionality of reflection and radiation from the earth’s surface, can affect the quality and reconstruction of orthophotos from UAV.

To avoid this problem, this work shows that, despite the low number of orthophoto reconstructions (15), significant results can be obtained for some parameters used in orthophoto reconstruction (i.e., % Camera stations/Nimg and Reprojection error) but also for other parameters used, even if they do not report significant results, which are still advantageous to the orthophotos from calibrated images. By increasing the number of images, and thus reconstructions, significant differences can be achieved. Therefore, this work contributes to improving the repeatability of drones’ survey image acquisition, allowing a more homogeneous and quantitatively valid orthophoto reconstruction using calibrated images. This allows for the reduction or the elimination of defects due to different lightning conditions through reconstruction and calibration.

The comparison of the calibrated and non-calibrated orthophotos through a Wilcoxon paired test on the 300 RMS SD of the three R, G, and B channels show a highly significant (*p* < 0.0001) difference between medians (calibrated = 5.7 ± 2.0; non-calibrated 8.0 ± 3.1).

As demonstrated by Menesatti et al. [17], the calibration algorithm 3D Thin-Plate Spline (TPS3D) obtained a colour reconstruction with an error lower than 4% (other standard methods applied by software showed errors ranging between 6 and 12%), allowing a quantitative measurement of the environmental lights and different devices used. Indeed, RGB acquired values are very device dependent. This leads to the possibility for the agricultural sector to carry out repeated measurements over time and quantify the differences relative to a specific study case.

Therefore, the measured RGB values show that the calibrated images have a statistically high difference in medians between calibrated and uncalibrated images, achieving greater homogeneity than the uncalibrated images, and, consequently, can also be used from a quantitative point of view [17].

In addition, the proposed colour calibration protocol could be useful when integrated into robotic and sensor platforms for the environmental exploration and monitoring of different environments from marine [22] to extraterrestrial [23].

## 5. Conclusions

UAVs play an important role in precision farming and many other application fields, accurately identifying problems or creating maps for precise agricultural field management. The benefits regard both agricultural and forestry sustainability. However, remote sensing via UAV is often influenced by several environmental factors including weather conditions and, in particular, the brightness during the different hours of the day. Moreover, the frequency rate of the illumination conditions is comparable with the inverse of the measurement timing (i.e., the time during which the measure is conducted). As a result, the noise due to the variability of the illumination conditions does not average to zero and cannot be reabsorbed into an illumination offset. Therefore, it is crucial to use a proper calibration method for a proper reconstruction. This colorimetric calibration was shown to be a potential tool to contain or eliminate the problems arising during the orthophotos’ reconstruction. Often, a flight performed during a precise phenological plant stage or, for example, during a pathogen attack which is crucial for an agrochemical intervention, is not repeatable. This tool could aid orthophoto use even when some problems arise during flight, lowering the final errors. The number of reconstructions used in the present work was low, however, from a future perspective, increasing their number would have a greater meaning and therefore a stronger calibration could be obtained along with the derived benefits. In conclusion, the results obtained can boost the performances of UAVs through the application of an innovative calibration algorithm. Moreover, the technique, producing more stable and standardised images in terms of light and colour, can broaden the environmental condition in which it is possible to fly, thus increasing the surveying potential of valid precision agriculture tools.

## Figures and Tables

**Figure 1 sensors-22-06490-f001:**
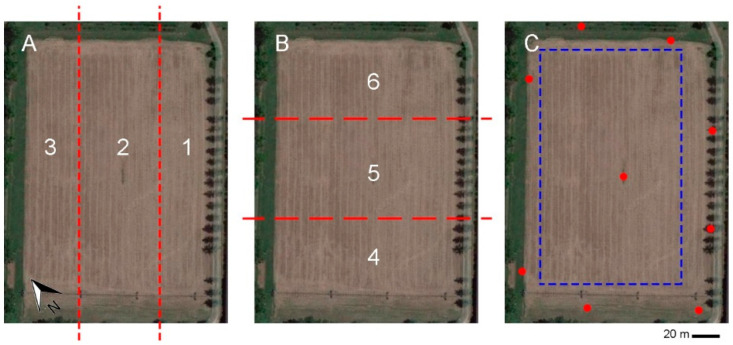
Setup of the experimental field. (**A**) Horizontal zones delimitated by red dashed lines (1-2-3); (**B**) longitudinal zones delimitated by red dashed lines (4-5-6); and (**C**) nine ground control points (red dots) and the central part of the agricultural field sampled with 300 random points (blue dashed rectangle).

**Figure 2 sensors-22-06490-f002:**
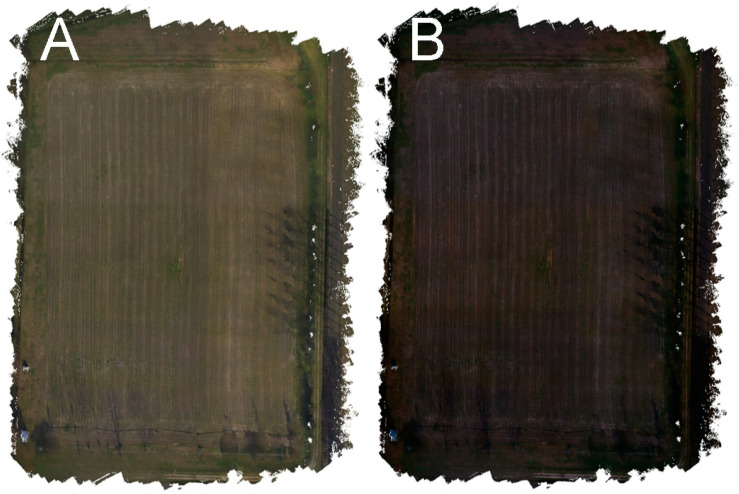
Examples of calibrated (**A**) and non-calibrated (**B**) orthophoto reconstructed randomly by combining the same six zones over the three days of the experiment.

**Figure 3 sensors-22-06490-f003:**
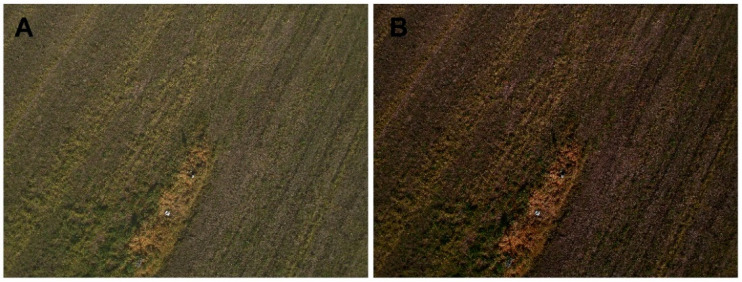
Examples of calibrated (**A**) and non-calibrated (**B**) images including a GCP; Day 2, Zone 5.

**Figure 4 sensors-22-06490-f004:**
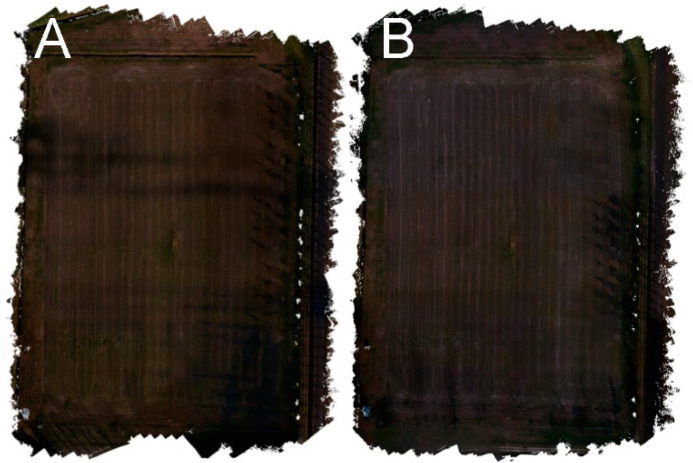
Examples of non-calibrated orthophoto reconstructed with images of the same day (day 1) (**A**), and randomly combining the same six zones over the three days of the experiment (**B**).

**Table 1 sensors-22-06490-t001:** Experimental setup: Weather conditions over the three days and daytimes of the flights over the six zones (Figure 1A,B).

Days	Zone 1	Zone 2	Zone 3	Zone 4	Zone 5	Zone 6
Day 1 (partly cloudy)	10.03	12.00	12.15	13.08	15.04	16.29
Day 2 (clear)	15.07	16.08	13.09	12.10	9.59	11.11
Day 3 (cloudy)	12.39	13.19	12.22	15.47	15.36	10.51

**Table 2 sensors-22-06490-t002:** Specifications of the unmanned aerial vehicle (UAV) DJI™ SPARK™.

Details	Items	Specifications
UAV	Weight	297 g
	Dimensions	143 mm × 143 mm × 55 mm
	Max speed	50 km/h
	Satellite positioning systems	GPS/GLONASS
Digital camera	Camera focal length	4.5 mm
	Sensor dimensions (WxH)	6.17 mm × 4.56 mm
	Sensor resolution	12 megapixels
	Image sensor Type	CMOS
	Capture formats	MP4 (MPEG-4 AVC/H.264)
	Still image formats	JPEG
	Video recorder resolutions	1920 × 1080 (1080 p)
	Frame rate	30 frames per second
	Still image resolutions	3968 × 2976
GIMBAL	Control range inclination	from −85° to 0°
	Stabilisation	Mechanical two axes (inclination, roll)
	Obstacle detection distance	0.2–5 m
	Operating environment	Surfaces with diffuse reflectivity (>20%) and dimensions greater than 20 × 20 cm (walls, trees, people, etc.)
Remote Control	Operating frequency	5.8 GHz
	Max operating distance	1.6 km
Battery	Supported batteryConfigurations	3S
	Rechargeable battery	Rechargeable
	Technology	lithium polymer
	Voltage provided	11.4 V
	Capacity	1480 mAh
	Run rime (up to)	16 min
	Recharge rime	52 min

**Table 3 sensors-22-06490-t003:** Mean (±SD) values of the parameters obtained after orthophoto reconstruction. Differences between the calibrated and non-calibrated reconstructions. Means with the same letter are not significantly different (Wilcoxon paired test).

Parameters	Calibrated	Non-Calibrated
% Camera stations/Nimg	99.97 ± 0.05 ^a^	92.50 ± 14.43 ^b^
Tie points	4,188,281 ± 62,345 ^a^	3,880,425 ± 63,0471.1 ^a^
Projections	11,465,475 ± 304,727 ^a^	10,650,366 ± 1,584,657 ^a^
Reprojection error	2.02 ± 0.70 ^a^	2.37 ± 1.22 ^b^
Control points RMSE Total	1.95 ± 2.28 ^a^	2.43 ± 2.93 ^a^
Check points RMSE total	12.42 ± 25.56 ^a^	15.49 ± 26.45 ^a^
RMS SD-RGB	6.0 ± 2.0 ^a^	8.7 ± 3.1 ^b^

## Data Availability

Not applicable.

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
