# Peer review of "Advantages in Using Colour Calibration for Orthophoto Reconstruction"

_sensors, 2022, doi:10.3390/s22176490_

Round 1

Reviewer 1 Report

In “Advantages in using colour calibration for orthophoto reconstruction” the authors explore the possibility to use a color calibration algorithm to improve the reconstruction of orthophotos. Spline interpolation algorithm was applied to pictures of a field taken in three different flight sessions under different weather conditions.

Thirty random reconstructions were obtained considering the 3 days and 6 different areas; these datasets were utilized to compare the calibrated and not calibrated images characteristics/parameters. Results show a general improvement of considered parameters and, in particular, significant differences in the percentage of camera stations on the total number of images and the reprojection error.

In the reviewer’s opinion, the manuscript is surely interesting and the results obtained can be of great help in the field, however, some points should be clarified to fully understand the benefits of the proposed approach. Here below is a list of points that, in the reviewer’s opinion, should be comments or taken into consideration.

1)     A brief description of parameters utilized to estimate the performance of the algorithm reported in table 3 should be provided.

2)     If the reviewer correctly understands, the errors in the color were calculated using GCP. Pictures of GCP in a real/standard image and in calibrated and not calibrated images should be provided to better understand the benefits of the algorithms and the results obtained. As well as, a real/standard image of the field should be of benefit.

3)     In the manuscript, the authors wrote: “A total of 30 orthophotos have been obtained. Fifteen from calibrated and fifteen from non-calibrated images among which three from the reconstruction of the zones of each day and twelve randomly combining the 6 zones over the three days of the experiment”. Even if the dataset is exiguous, it is possible to see or appreciate a difference in the two cases (same day – different days)?

4)     Is there a criterion for the choice of the Ground and Quality Control Points? Are they always the same or they have been chosen randomly each time? May this choice influence the results?

5)     Since it is the main point of the work, a short explanation of the thin-plate spline interpolation function should be provided, in spite of the presence of the reference.

6)     In the text, the author wrote “Before the flight, an image color checker GretagMacbeth (24 patches) (X-Rite, Michi-152 gan, United States) was acquired for a posteriori color calibration.” It is not clear the role of these images and the meaning of posteriori calibration.

7)     Authors utilized GPC as the acronym for Ground Control Points. Should not it be GCP, instead?  

8)     Finally a curiosity. Is there a difference in the performance or errors in the three different RGB channels? Is there a channel more affected by weather/external illumination conditions?

Author Response

Reply to Reviewer #1

In “Advantages in using colour calibration for orthophoto reconstruction” the authors explore the possibility to use a color calibration algorithm to improve the reconstruction of orthophotos. Spline interpolation algorithm was applied to pictures of a field taken in three different flight sessions under different weather conditions.

Thirty random reconstructions were obtained considering the 3 days and 6 different areas; these datasets were utilized to compare the calibrated and not calibrated images characteristics/parameters. Results show a general improvement of considered parameters and, in particular, significant differences in the percentage of camera stations on the total number of images and the reprojection error.

In the reviewer’s opinion, the manuscript is surely interesting and the results obtained can be of great help in the field, however, some points should be clarified to fully understand the benefits of the proposed approach. Here below is a list of points that, in the reviewer’s opinion, should be comments or taken into consideration.

1) A brief description of parameters utilized to estimate the performance of the algorithm reported in table 3 should be provided.

As suggested, a brief description of parameters utilized to estimate the performance of the algorithm has been added in M&M section.

2) If the reviewer correctly understands, the errors in the color were calculated using GCP. Pictures of GCP in a real/standard image and in calibrated and not calibrated images should be provided to better understand the benefits of the algorithms and the results obtained. As well as, a real/standard image of the field should be of benefit.

No, errors in colors (or, better, the homogeneity of the colors) has been measured using the 300 values of the RMS standard deviations (SD) associated with the three channels of the calibrated and non-calibrated orthophoto. We better and extensively explain this result in the text and, as requested we added a Figure.

3) In the manuscript, the authors wrote: “A total of 30 orthophotos have been obtained. Fifteen from calibrated and fifteen from non-calibrated images among which three from the reconstruction of the zones of each day and twelve randomly combining the 6 zones over the three days of the experiment”. Even if the dataset is exiguous, it is possible to see or appreciate a difference in the two cases (same day – different days)?

As requested, we added a new Figure.

4) Is there a criterion for the choice of the Ground and Quality Control Points? Are they always the same or they have been chosen randomly each time? May this choice influence the results?

GCP and checkpoint are always the same for all the calibrated and non-calibrated re-constructions in order to avoid influences on the results. We added this explanation in the M&M section.

5) Since it is the main point of the work, a short explanation of the thin-plate spline interpolation function should be provided, in spite of the presence of the reference.

As suggested, we added information about the thin-plate spline interpolation function related to Menesatti et al. [17].

6) In the text, the author wrote “Before the flight, an image color checker GretagMacbeth (24 patches) (X-Rite, Michi-152 gan, United States) was acquired for a posteriori color calibration.” It is not clear the role of these images and the meaning of posteriori calibration.

We better explain this part. At the beginning of each flight, an image of the color checker has been acquired. This image serves for the a posteriori color calibration.

7) Authors utilized GPC as the acronym for Ground Control Points. Should not it be GCP, instead?

As suggested, we replaced the acronym “GPC” with “GCP”.

8) Finally a curiosity. Is there a difference in the performance or errors in the three different RGB channels? Is there a channel more affected by weather/external illumination conditions?

The RMS has been calculated on the standard deviations in the three R, G and B channels. We do not measure differences considering the three channels for separate. In our opinion, in the RGB space, the three channels could result similar, maybe using the Lab color space the L channel could have a stronger effect due to the different weather and daytime conditions.

Reviewer 2 Report

Dear Authors,

Please find below the comments and suggestions:

1. Follow the Journal guideline to build Abstract. It seems to be adjusted for other one.

2. This work is under Physical sensor section. However, the work does not shows any physical concepts.

3. The manuscript is called as Article, however, it seems to be a technical note. The length of the manuscript is very short and lacks an appropriate description of the methodology, results obtained and main contributions in the area.

4. Conventional software and techniques are used in the calibration protocol, so what is the contribution? A new module, algorithm, etc.

5. As the experiment just takes 3 days, the authors can increase the experiment locations and samples to get a proper statistical description. Fig. 2 does not shows a critical improvement, in particular, for non-proficient readers.

6. The improvement of Images can be evidenced by using a density color plot and 2D curve plots as Ref. 15.

7. Conclusions must be improved. Just line 259-260 tries to give light of the findings obtained.

Author Response

Reply to Reviewer #2

Dear Authors,

Please find below the comments and suggestions:

  1. Follow the Journal guideline to build Abstract. It seems to be adjusted for other one.

The Abstract has been modified.

  1. This work is under Physical sensor section. However, the work does not shows any physical concepts.

The meaning of the submission in the Physical sensor section is the use of an RGB sensor on a light-drone commercial drone. The physical unexpensive sensor coupled with a complex (but easy to-use) color calibration procedure are able to better reconstruct orthophoto. Indeed, the frequency rate of the illumination conditions is comparable with the inverse of the measurement timing (i.e., the time during which the measure is conducted). As a result, the noise due to the variability of the illumination conditions does not average to zero and cannot be reabsorbed into an illumination offset. Therefore, it is crucial to use a proper calibration method for proper reconstruction. We added this consideration in the Conclusion section.

  1. The manuscript is called as Article, however, it seems to be a technical note. The length of the manuscript is very short and lacks an appropriate description of the methodology, results obtained and main contributions in the area.

As also suggested by Rev #1 we added more explanations and some more figures.

  1. Conventional software and techniques are used in the calibration protocol, so what is the contribution? A new module, algorithm, etc.

The paper used a color calibration algorithm (non-conventional) applied to drone images to evaluate the differential performances using calibrated and non-calibrated images. This is new, as also observed by rev #1, in the use of drones in the precision agriculture framework.

  1. As the experiment just takes 3 days, the authors can increase the experiment locations and samples to get a proper statistical description. Fig. 2 does not shows a critical improvement, in particular, for non-proficient readers.

We decided to work in the cold season on three consecutive days in order to avoid interferences due to plant growth. The principles obtained by this article could serve in many different applicative fields, from agriculture to forestry, where repeated measures are necessary.

  1. The improvement of Images can be evidenced by using a density color plot and 2D curve plots as Ref. 15.

Ref. 15 used a specific radiometric NIR device. In our work we estimate the color homogeneity by considering the Root Mean Square (RMS) of the standard deviations in the three R, G and B channels. As also requested by Rev #1 we better explain this result.

  1. Conclusions must be improved. Just line 259-260 tries to give light of the findings obtained.

The conclusion section has been improved.

Round 2

Reviewer 1 Report

the authors satisfactorily commented and amended the manuscript according to the reviewer's comment. In the reviewer's opinion, the manuscript is worth to be published in the present form.

Reviewer 2 Report

Thank you for the revised version. 

I recommend the publication of the work.